

# The impact of COVID-19 pandemic on nitrogen dioxide levels in Nigeria

Johnson Adedeji Olusola[1], Adebola Adekunle Shote[2], Abdellah Ouigmane[3,4] and Rima J. Isaifan[5]

[1] Institute of Ecology and Environmental Studies, Obafemi Awolowo University, Ile Ife, Osun, Nigeria
[2] GIS Unit, Ogun State Water Corporation, Abeokuta, Ogun State, Nigeria
[3] Applied Spectro-Chemometry and Environment Department, University of Sultan Moulay Slimane, Beni Mellal, Morocco
[4] Agro-Industrial and Environmental Processes Department, University of Sultan of Moulay Slimane, Beni Mellal, Morocco
[5] Division of Sustainable Development, College of Science and Engineering, Hamad Bin Khalifa University, Qatar Foundation, Education City, Doha, Qatar

Corresponding authors
Johnson Adedeji Olusola,
johnson.olusola@eksu.edu.ng
Rima J. Isaifan, risaifan@hbku.edu.qa

## ABSTRACT

The Coronavirus disease (COVID-19) has been transmitted worldwide over a very short time after it originated in China in December 2019. In an attempt to control its spread and reduce its health impacts, several countries including those in the African continent imposed restrictive measures that was termed "lockdown". The outcomes of this lockdown have been reported to be beneficial to air quality worldwide. The main objective of this study is to assess the impact of lockdown due to COVID-19 on nitrogen dioxide ($NO_2$) levels over six major cities in Nigeria. Maps extracted from  satellite (Sentinel-5P) were used to indicate the significant reduction in the level of $NO_2$ in the selected cities in Nigeria during two time-intervals, pre-lockdown (December, 2019) and during lockdown (April, 2020). The results show a significant reduction in $NO_2$ levels during the lockdown period compared with its levels during the pre-lockdown period in 2019. The reduction in $NO_2$ concentration levels during lockdown is likely due to less traffic, social distancing and restrictions on business and human activities. There could be an element of uncertainty in the results due to seasonality, as the comparison is done with a different season. However, the magnitude of change due to lockdown is probably much higher than the seasonal variability.   Although COVID-19 has negatively impacted the health and economic status of all regions worldwide, it has benefited some aspects of air quality in most countries including Nigeria. This indicates that anthropogenic activities may be managed to reduce air pollution and positively impact the health of human beings.

## INTRODUCTION

The novel coronavirus (COVID-19) has changed human behavior and resulted in hundreds of thousands of deaths, caused excessive loss of jobs, and reduced social and economic activities. The first confirmed case of this deadly virus was seen in Wuhan

province in China in December, 2019 before it spread across the globe and was declared by the World Health Organization (WHO) as a pandemic in March 2020 (*Isaifan, 2020*). Severe Acute Respiratory Syndrome Coronavirus-2 (SARS CoV-2) has been confirmed as the pathogen responsible for COVID-19 (*Zhoua et al., 2020*) which had previously caused Severe Acute Respiratory Syndrome (SARS) (*Peiris, Guan & Yuen, 2004*) and Middle East Respiratory Syndrome (MERS) (*Zaki et al., 2012*).

Nigeria reported the first case of COVID-19 on the 27[th] of February 2020, following the arrival of an Italian citizen in Lagos from Europe (*Kalu, 2020*). Due to contact with the first case, a second confirmed case was reported in Ogun State, and then more cases quickly emerged (*Kalu, 2020*). Eventually, COVID-19 spread to all the States in the Federal Republic of Nigeria as reported by the Nigerian Center for Diseases Control (NCDC) which showed that all the states in Nigeria had one or more cases, with a confirmed fatality rate (CFR) of 3%. Accordingly, the Federal Government of Nigeria formed the Presidential Task Force (PTF), to oversee the pandemic. In the course of this action, a 2-week lockdown was announced by the Nigerian Government on March 30[th], 2020, in three states; Lagos, Ogun and Abuja, the first states hit by the virus before the lockdown was further extended to the whole country. Afterwards, there have been a series of extensions of this lockdown by both Federal and State Governments of Nigeria. Recently, the lockdown has been gradually eased in phases to allow basic activities and normal business. The World Health Organization (WHO) believes that lessons learnt from the Ebola crisis in 2014 in Africa have improved the resilience of the African healthcare systems and their capacity to test and successfully impose quarantine (*Fox, 2020*). Countries across the continent have garnered a great deal of experience from tackling infectious diseases like polio, measles, Ebola, yellow fever, influenza and many more (*Bruce-Lockhart, 2020*). Hence, the African governments were quick to impose travel bans on airlines carrying passengers from Europe at the very beginning of the pandemic while Europe and North American countries went into self-imposed economic and social lockdown at a later stage. Economically, the UNECA reckons that Nigeria could suffer a $19 billion hit, largely from lost oil revenues (*Fox, 2020*).

During the long lockdown in Nigeria, many economic and social activities were brought to a halt; commercial activities; vehicular movements (private and governmental owned); most industrial activities, especially in the oil rich cities of the Delta which account for the greater percentage of emissions in Nigeria, and other human activities which invariably reduced anthropogenic emissions. Air pollution (release of harmful substance to the atmosphere in a concentration that could pose serious threats to humans, animals and plants) is a serious concern to the world. Air quality can be affected by natural sources such as volcanic eruptions, and natural disasters, and by anthropogenic sources such as the combustion of fossil fuels (diesel, gasoline, and oil) used for energy production and by industrial activities related to construction, mining, cement manufacturing, and smelting (*Li & Mallat, 2018*; *Al-Thani, Koç & Isaifan, 2018*; *Al-Thani et al., 2020*).

Several studies have been carried out to assess the impacts of air pollution on humans as it poses a serious health risk. For instance, the exposure to an elevated amount of $NO_2$ has been associated with hypertension illness (*Shin et al., 2020*). Several studies have shown

an increased rate of heart-related diseases due to air pollution exposure (*Li et al., 2016*; *Logue et al., 2012*). *Nwachukwu, Chukwuocha & Igbudu (2012)* reported on the effect of air pollution on inducing several diseases for the people living in the vicinity of Rivers State in Nigeria by comparing epidemiological data to air quality data. The results showed that a total of 30,435 cases of illness (e.g., pertussis, pulmonary tuberculosis, cerebrospinal meningitis (CSM), pneumonia, measles, chronic bronchitis, and upper respiratory tract infection (URT)) were reported between 2003 and 2008, of which 61 patients died.

There have been reports on the reduction of air pollution since the advent of coronavirus in various cities of the world. Isaifan has initially reported on the air quality status in China before and after the coronavirus crisis (*Isaifan, 2020*). It has been shown that shutting down industrial activities which are considered intense sources of air pollution, in response to the COVID-19 pandemic emergency lockdown may have saved more lives by preventing air pollution than by preventing infection with the virus. In India, *Gupta, Tomar & Kumar (2020)* analyzed various environmental impacts due to the lockdown as prevention for COVID-19 throughout India. They observed a significant reduction in the main air pollutants such as nitrogen dioxide, carbon monoxide, sulphur dioxide, ozone and particulate matter, hence the positive impact of containment of air quality. In India, *Sharma et al. (2020)* investigated the impact of COVID-19 on restricting activities by analyzing the concentrations of six criteria pollutants, $PM_{10}$, $PM_{2.5}$, CO, $NO_2$, ozone and $SO_2$ during March 16th to April 14th from 2017 to 2020 in 22 cities covering different regions of India. They reported an overall 43, 31, 10, and 18% decreases in $PM_{2.5}$, $PM_{10}$, CO, and $NO_2$; respectively in India during lockdown period compared to previous years while, there were 17% increase in $O_3$ and negligible changes in $SO_2$.

Using ground-based monitoring stations, IQAir Air Visual (2019) World air quality assessed the impact of COVID-19 restrictions on $PM_{2.5}$ levels for 10 major cities around the world. The reduction in $PM_{2.5}$ was observed in nine of the 10 cities compared to the same period in 2019. The best air quality was recorded in Wuhan in February and March 2020. Delhi air quality dropped from 68% in 2019 before the lockdown period to 17% during the lockdown period, while the longest period of clean air was recorded in Los Angeles in compliance with WHO air quality guidelines.

The study reported here explored the implications of the lockdown due to COVID-19 on Nigerians' air quality, and more specifically on $NO_2$ levels. The chosen cities are Lagos, Ibadan, Abuja, Kano, Owerri and Onitsha which are distributed along different locations in the country. In addition, these cities are among the top major inhabited cities in Nigeria based on their demography.

## Lagos and Ibadan

Lagos State lies in the southwestern part of Nigeria within latitudes 6° 23′ N and 6° 41′ N and longitudes 2° 42′ E and 3°42′ E. Lagos is the most populous city in Nigeria and in all Africa, and is considered the financial center of the metropolis. Fifty percent of Nigeria's industrial activities including 300 industries in 12 industrial Estates are located in the Lagos area. The high urbanization and industrial growth rate in Lagos has made it one of the most densely-populated regions on earth with about 9.3 million inhabitants according

to 2006 Census (*Adesuyi, Njoku & Akinola, 2015*). The city has a tropical climate with an average relative humidity of 79%. The mean monthly temperature ranges from 23–32 °C. Being located in a coastal area and influenced by strong sea based disturbances, Lagos experiences an average wind speed of 4.3 km/h (*Komolafe et al., 2014*). Lagos metropolis has been experiencing air pollution problems in all its severity over the past decades (*Njoku et al., 2016*). Sources of air pollution include traffic (vehicle exhaust), industrial sectors (from brick making to oil and gas production), power plants and generating sets, cooking and heating with solid fuels (e.g. coal, wood, crop waste), forest fires and open burning of municipal waste and agricultural residues (*Akanni, 2010*; *Komolafe et al., 2014*; *Lawal, 2004*).

Ibadan is the capital of Oyo State, South West Nigeria. Ibadan is positioned on longitude 3°53E of Greenwich Meridian and latitude 7°23N of the equator. This ancient city is located close to forest and grassland boundary of south western Nigeria and about 145 km North East. Due to its location, Ibadan serves as a meeting point for people and products from forest and grassland areas (*Rowland, 2015*). The city of Ibadan is known to be the third largest metropolitan area in Nigeria after Lagos and Kano. This is because it is one of the fastest urbanizing cities in Nigeria. The increase in urbanization is attributed to the provision of better economic opportunities due to setting up of factories and industries, which has led to migration of population from rural regions to the city. As a result, people spread to the peripheral areas of the urban fringes (*Rowland, 2015*; *Owoeye & Ogundiran, 2015*).

## Kano

Kano is the capital of Kano State, Nigeria which lies between latitude 11° 55′ 23.93′ N to 12° 3′ 53.10′ N and longitude 8° 27′42.26′E to 8° 36′41.62′E and is 1,549 feet above sea level. The estimated area of Kano metropolis increased from 122.7 square kilometers in 1962 to 154.6 square kilometers in 1981, an increase of about 25% based on the average expansion rate of two square kilometers per annum (*Na'Abba, 2002*).

Kano is referred to as the Center of Commerce in the Country due to its long flourished marketing activities. This is based on the fact that marketing and trading has been the dominant economic activities in Kano. However, the land is mostly exploited by urban agriculture through waste water utilization to sustain daily needs. The average temperature is a bit hot, even during the cool Harmattan period, the minimum temperature hardly falls below 11 °C, whilst the monthly average temperature is not less than 20 °C. During the hot season, usually Mid-March to Mid-May, the maximum temperature reading can reach as high as 40 °C. The average temperature for these hot months may range between 30 °C and 32 °C (*Adzandeh, Fabiyi & Bello, 2014*).

## Abuja

Abuja is the federal capital of Nigeria and is located at 6°45″ longitude and 7°45″ E from Greenwich and at 8°25″ latitude and 9°25″ N from the equator. It is bounded on the north by Kaduna State, on the east by Nassarawa State, on the west by Niger State and on the south by Kogi State. The topography of the region is characterized by the presence of several peaks which constitute the highest part of the region; located at 760 m above sea

level (*Balogun, 2000*). Climatically, the region has a high intensity of precipitation which reaches 1,631.7 mm per year and an average annual temperature between 25.8 °C and 30.2 °C. Abuja has an international airport and is linked to other Nigerian cities by a network of highways.

### Owerri and Onitsha

Owerri is the capital of Imo State, located in south-eastern Nigeria. It is bounded on the south by Nekede, on the north by Amakohia, on the east by Egbu, on the northeast by Uratta, on the southeast by Naze, and on the northwest by Irette. The main activities of the inhabitants are public functions, commerce and agriculture (*Obionu, 2007*). The average temperature is around 27 °C (*Obionu, 2007*). The vegetation is typically tropical rainforest with some parts of Guinean savannah as a result of environmental degradation through pollution.

The city of Onitsha is located on a vast flood plain on an altitude of 26 m above sea level between latitudes 6°5′ to 6°11′N and longitudes 6°45′ to 6°53′E. The climate type is equatorial tropical with dry and wet seasons. The rainy season extends from March to October and the dry season begins from November to February. The region is located in the rainforest belt, with an average rainfall of 2,100 mm. According to the World Bank, Onitsha is the most polluted city in the world for air quality as measured by the concentration of small particles ($PM_{10}$). The main factors contributing to air pollution are waste combustion, emissions from old cars and the heavy use of fossil fuels for cooking (*World Health Organization, 2016*).

The combustion of fossil fuels in these selected cities generates large quantities of nitrogen dioxide ($NO_2$), but traffic is considered the main source. $NO_2$ causes cellular inflammation, bronchial hyper-reactivity and respiratory problems, reportedly causing the death of about 4.6 million people each year (*Muhammad, Long & Salman, 2020*; *Latza, Gerdes & Baur, 2009*). To investigate the air quality impact of the lockdown due to COVID-19, each city's tropospheric column of nitrogen dioxide was analyzed. This study compares the levels of $NO_2$ density before the advents of the first reported case of COVID-19 in Nigeria and its levels during the lockdown in selected cities to assess its impact on the air quality of Nigeria.

## MATERIALS & METHODS

For $NO_2$ concentration in the troposphere (from surface up to ~10 km), the Sentinel-5 Precursor space-borne satellite (spatial resolution of 5.5 km) was used which is operated and managed by the European Commission under the "Copernicus" program. The satellite operates in a sun-synchronous orbit at 824 km and an orbital cycle of 16 days. The satellite carries a TROPOspheric Monitoring Instrument (TROPOMI) which provides a (near-) global coverage of air pollution caused by $NO_2$ and other pollutants such as $O_3$, $SO_2$, CO, $CH_4$, and aerosols. (*Veefkind et al., 2012*).

This study necessitated the extraction of $NO_2$ data from Sentinel 5P Near Real Time level 2 satellite images achieved in NetCDF format and captured for specific exposure durations in 2 separate months (December 2019 and April 2020). From the sentinel 5P

**Table 1 Data type and sources of data collected for this research.**

| No. | Name type | Data source | Date acquired | File name | Filename extension |
|---|---|---|---|---|---|
| 1 | Nitrogen dioxide tropospheric column | http://s5phub.copernicus.eu/dhus/#/home | April 2019, December 2019, April 2020 | NetCDF | .nc |
| 2 | National and State Boundary | http://diva-gis.org/data | March, 2020 | Shape file (Polygon) | .shp |
| 3 | State Capital | Google Earth Pro | March, 2020 | Keyhole Markup Language Zipped | Kmz/Kml |

satellite images, the extracted $NO_2$ data were used to compare the trend of $NO_2$ spread in the atmosphere before and during the COVID-19 lockdown in the Nigerian cities selected (Lagos, Owerri & Onitsha, Abuja and Kano).

The Panoply 4.11.2 application was used to process the Network Common Data Format files (NetCDF) to raster images using two dimensions latitude and longitude (in that order) and scaled to reflect the proper $NO_2$ concentration, ArcMap 10.5 was used to analyze the $NO_2$ sentinel 5P satellite images, projecting then to the scale of the selected cities, likewise, applying standard deviation stretch to address the image stretch property as a result of the analysis carried out on the images.

For the secondary data which includes Sentinel 5P $NO_2$ Satellite imagery (Table 1) covering the study area in NetCDF file format were obtained from Copernicus website and the Tropospheric column of Nitrogen dioxide file for the months of April and December was used in this research work.

In addition, administrative boundaries (National and State boundary shape files) of Nigeria was sourced from DIVA-GIS website a repository of free spatial data. The resultant image provided information on $NO_2$ concentrations within the study areas. The maps were scaled to 1:60,000.

Additional secondary data were also obtained from a number of literatures ranging from text books, research articles and journal papers. Table 1 shows the various data types used in this research and their sources.

Sentinel 5P Near Real Time Level 2 images for the months of December 2019 and April 2020 for the study area, were plotted in Panoply. The tropospheric vertical column of nitrogen dioxide of each NetCDF file was plotted using the georeferenced longitude and latitude color contour plot. Using the scalar unit of micromole per meter square with a scale range minimum of 0 and maximum of 100, the plotted images were later exported to tagged image file format for further analysis in ArcMap.

The Tagged Image file (TIF) plotted in Panoply environment was processed in ArcMap, the image was georeferenced and projected to UTM 31; clipped using the Nigeria National Boundary; overlaid with the State boundary and State capitals.

## RESULTS

Figure 1 shows the map of Nigeria and highlights the areas included in this study. Table 2 shows the main features of the cities chosen for this study. Figure 2 shows the demographic distribution of confirmed cases in Nigeria. It can be seen that highly dense and

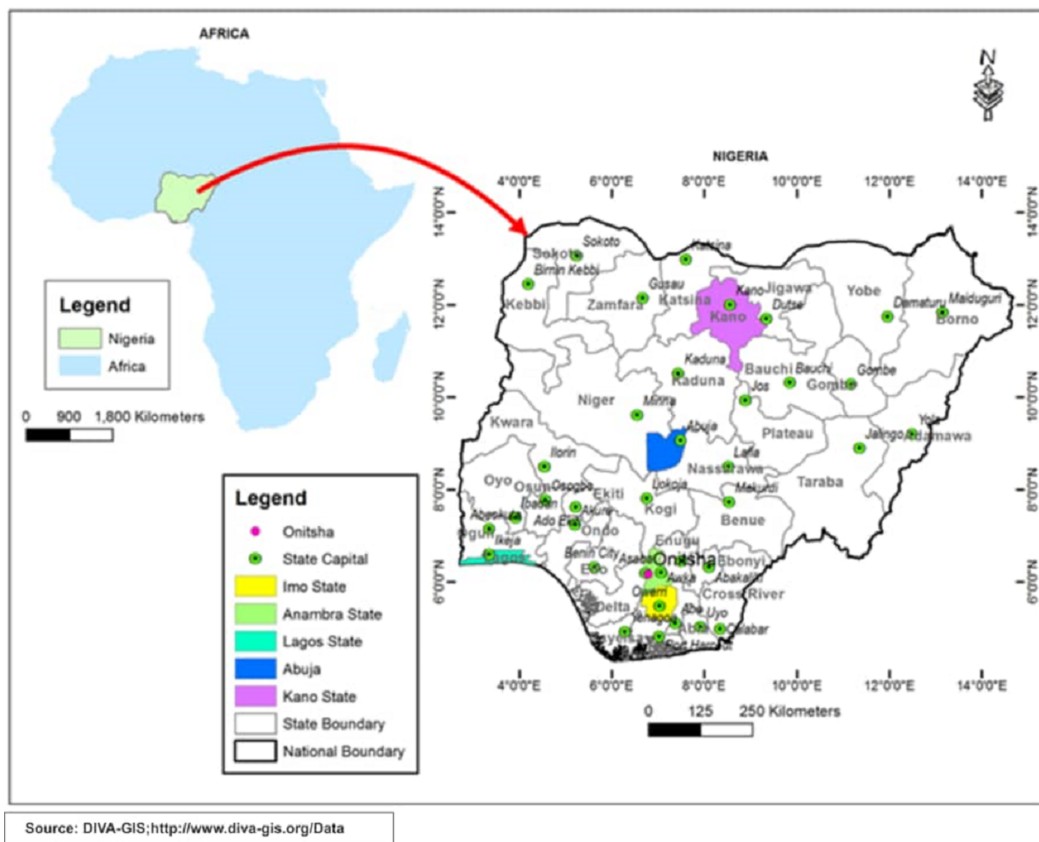

**Figure 1 Map showing Nigeria and areas that have been included in this study.**

**Table 2 Population, area, and population density in each of the cities included in this work as per Worldometer census data.**

| City | Population (2020) | Area (km²) | Population density (person/km²) |
|------|------------------|-----------|-------------------------------|
| Lagos | 9,000,000 | 1,171 | 7,685 |
| Kano | 3,626,068 | 449 | 8,076 |
| Ibadan | 3,565,108 | 3,080 | 1,158 |
| Abuja | 590,000 | 1,769 | 334 |
| Owerri | 215,038 | 551 | 390 |
| Onitsha | 561,066 | 52 | 10,789 |

populated areas have witnessed more infections than other regions as expected. Moreover, the demographic distribution of infected cases show that male confirmed cases were dominant over female's with 9,117 cases for male and 4,347 female which accounted to 68% and 32%; respectively (Table 3 and Fig. 3). Moreover, Fig. 3 reveals that the most affected age group is 31 – 40 which makes up about 24% of all age-group cases.

Table 4 shows the total registered cases with COVID-19 globally and the new cases in the last 24 h as per the 10th of June, 2020. It can be seen that Africa has the least number of cases with 2.03% out of all registered cases globally. The same can be seen from the
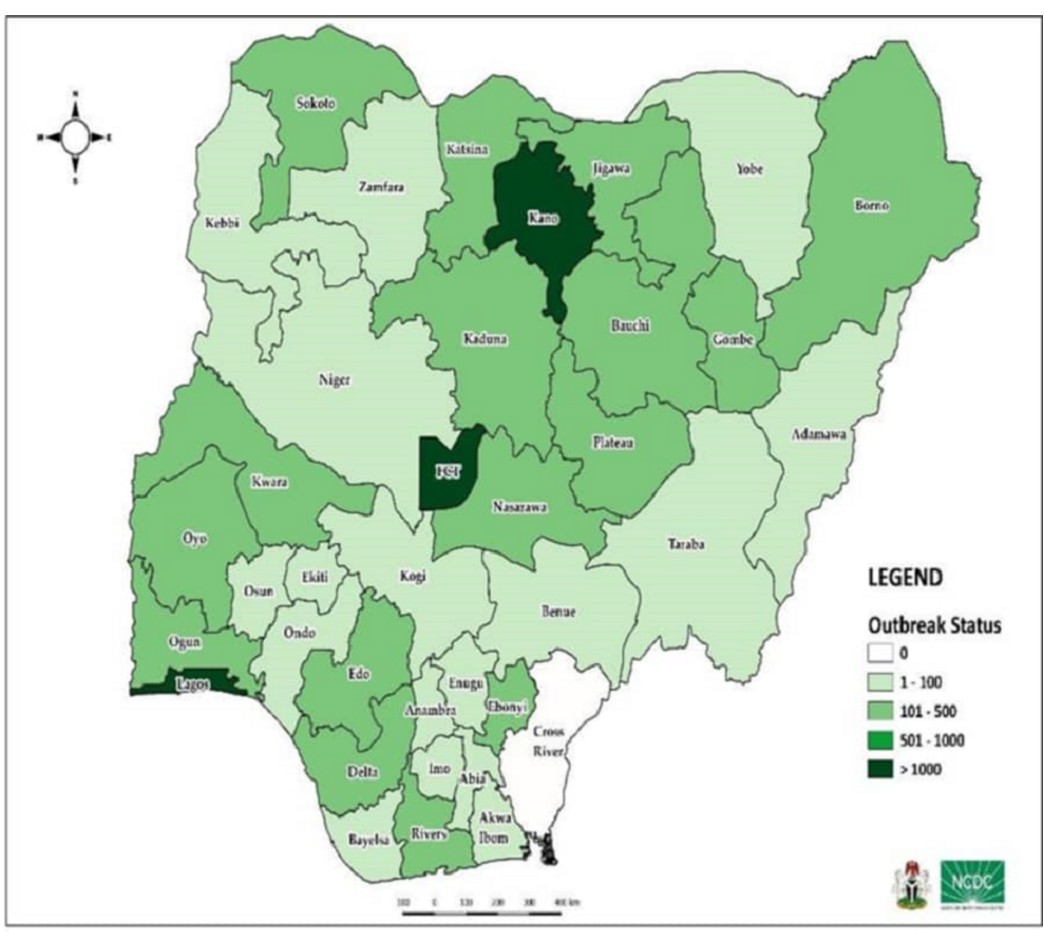

Source: Nigeria Centre for Disease Control (NCDC) 2020.

**Figure 2 COVID-19 outbreak status in Nigeria.** Source: Nigerian Centre for Diseases Control (NCDC) data as of June 10[th], 2020.

**Table 3 Demography and most affected age group by June 10[th], 2020.**

| Demography | | Most affected age group |
|---|---|---|
| Male | Female | |
| 9,117 (68%) | 4,347 (32%) | 31–40 (24%) |

number of deaths which amounted to 0.86% of global deaths by COVID-19. Moreover, Fig. 4 shows the epidemic curve of confirmed cases in Nigeria. In addition, figures in Table 5 shows Situation updates in numbers with the total cases (new cases in the last 24 hours) by June 10[th], 2020.

From the analysis of the map in Fig. 5B, it can be seen that nitrogen dioxide pollution has been widely present around the city of Lagos and its surroundings with high concentrations reaching 156.9 μmol/m$^2$ and above 575.7 μmol/m$^2$, with Ikeja the capital of the city recording 505.9305 μmol/m$^2$. Oyo State generally has a much lower nitrogen dioxide emissions between −52.36 μmol/m$^2$ to 471 μmol/m$^2$ compared to Lagos state,

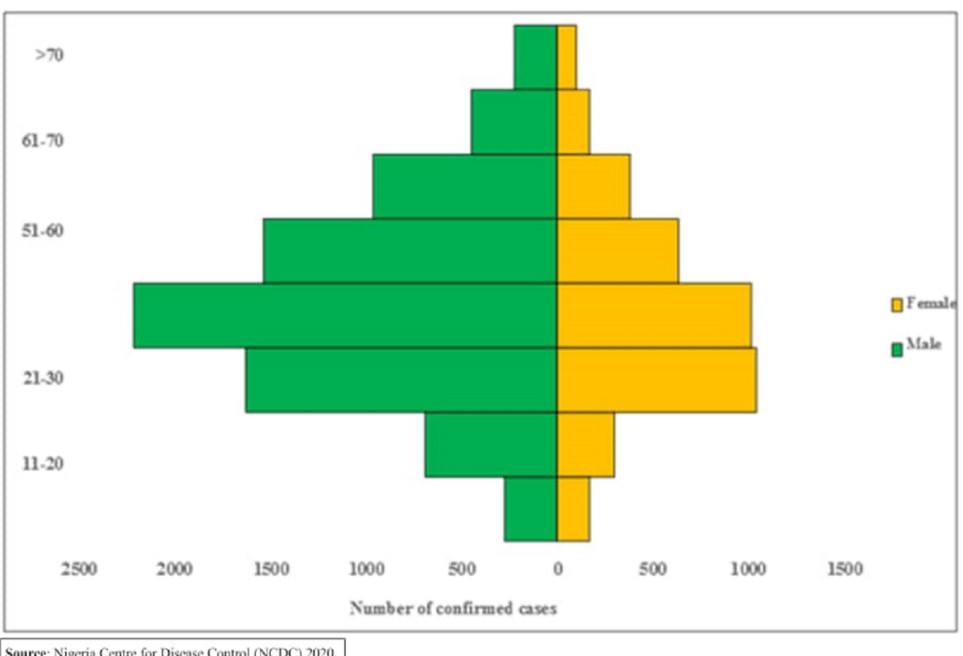

Source: Nigeria Centre for Disease Control (NCDC) 2020.

**Figure 3 Age-Sex distribution of confirmed cases.**

**Table 4 Situation in numbers by WHO Region.** Total positive cases (new cases in the last 24 h), and total deaths (new cases in the last 24 h) by June 10th, 2020.

| **Globally** | **7,145,539** | **cases (105,621)** | **408,025** | **deaths (3,629)** |
|---|---|---|---|---|
| Africa | 145,287 | cases (4,789) | 3,493 | deaths (141) |
| Americas | 3,415,174 | cases (48,923) | 185,863 | deaths (1,913) |
| Eastern Mediterranean | 677,338 | cases (18,724) | 15,246 | deaths (333) |
| Europe | 2,321,147 | cases (17,786) | 185,537 | deaths (866) |
| South-East Asia | 392,674 | cases (14,556) | 10,741 | deaths (365) |
| Western Pacific | 193,178 | cases (843) | 7,132 | deaths (7,132) |

**Note:**
Source: Situation Report–142 Data as reported by WHO from National Authorities by 10:00 CEST, by June 10th, 2020.

while Ibadan town has a concentration of 453.58 $\mu mol/m^2$ at the time before lockdown. Nevertheless, the map representing lockdown period (Fig. 5A) shows a visible decrease in nitrogen dioxide concentrations, with Ikeja measuring a negative value of $-60.41$ $\mu mol/m^2$ representing a decrease in Nitrogen dioxide concentration. Lagos overall concentration is between $-95.15$ $\mu mol/m^2$ and $-25.05$ $\mu mol/m^2$. This indicated significant decrease in activities that generate $NO_2$ due to lockdown restrictions.

The distribution of nitrogen pollution in the city of Kano is less severe compared to Lagos, only Kano town and its vicinity recorded a high concentration range between 471.03 $\mu mol/m^2$ and 366.34 $\mu mol/m^2$ while the rest have recorded values between $-50.46$ $\mu mol/m^2$ and 191.84 $\mu mol/m^2$ before lockdown as shown in Fig. 6B which is representative of December 2019. Whereas, during the containment period April 2020 (Fig. 6A), $NO_2$ pollution has reduced drastically in the entire Kano area, with values

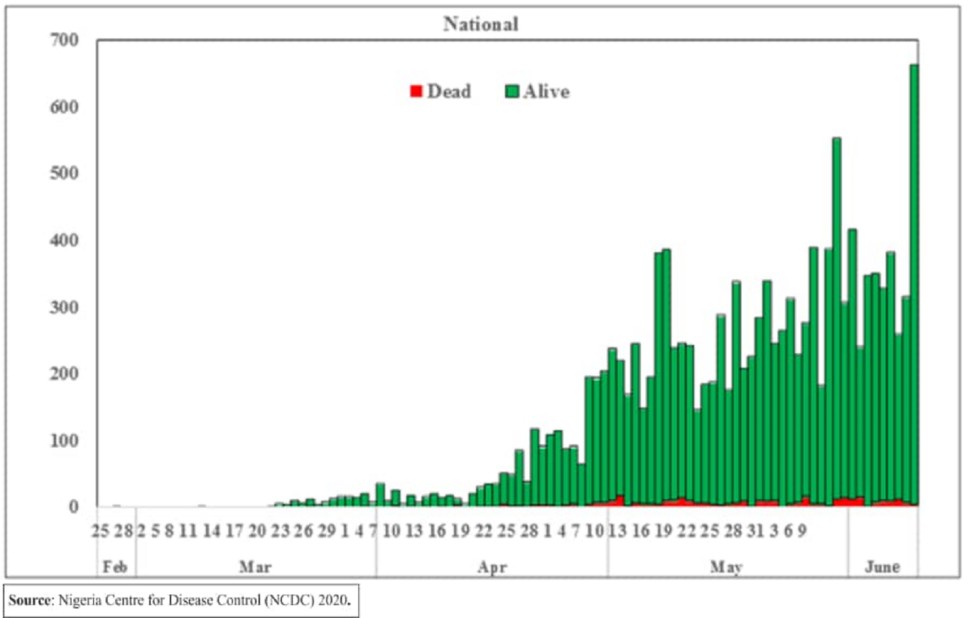

Source: Nigeria Centre for Disease Control (NCDC) 2020.

**Figure 4 Daily epidemic curve of confirmed cases.**

**Table 5 Situation updates in numbers.** Total positive cases (new cases in the last 24 hours) by June 10[th], 2020.

| Sample tested | Confirmed cases | Affected states | Discharged cases | Confirmed fatalities |
|---|---|---|---|---|
| 82,935 (2,987) | 13,464 (663) | 36 (0) | 4,206 (166) | 365 (4) 3% CFR |

ranging between −49.59 μmol/m$^2$ to 55.70 μmol/m$^2$ while Kano town records values at about −37.47 μmol/m$^2$.

Figure 7B shows that $NO_2$ has high concentrations in the city of Abuja and in the north of the city with the presence of several spots with high $NO_2$ concentrations during December 2019. Abuja city and northwest of Abuja city area show $NO_2$ concentration values ranging between 296.54 μmol/m$^2$ to 575.73 μmol/m$^2$. Other neighbouring towns southward Abuja marked values between 17.35 μmol/m$^2$ to 157 μmol/m$^2$. On the other hand, the results in Fig. 7A do not show any spots with $NO_2$ pollution due to lockdown measures, therefore accounting for a low $NO_2$ reading between −3.53 μmol/m$^2$ and 33.19 μmol/m$^2$ within the Federal Capital Territory, while Abuja Town accounts for readings around 26.23 μmol/m$^2$.

Based on the analysis of the results presented in Fig. 8, it is found that Onitsha area is the most polluted with the presence of several scattered spots representing $NO_2$ pollution throughout the region (Fig. 8B). The atmosphere in Onitsha town is charged with about 296.5 μmol/m$^2$ while Owerri town measures 471.03 μmol/m$^2$, other places measure as high as 156.9 μmol/m$^2$. Figure 8A shows a significant drop in $NO_2$ concentrations during the lockdown period. Onitsha records 15.94 μmol/m$^2$ (94%) reduction while other places within the state measure 35.01 μmol/m$^2$ (highest within the state) while Owerri town records 38.11 μmol/m$^2$, with lowest value of −18.17 μmol/m$^2$ and highest value of

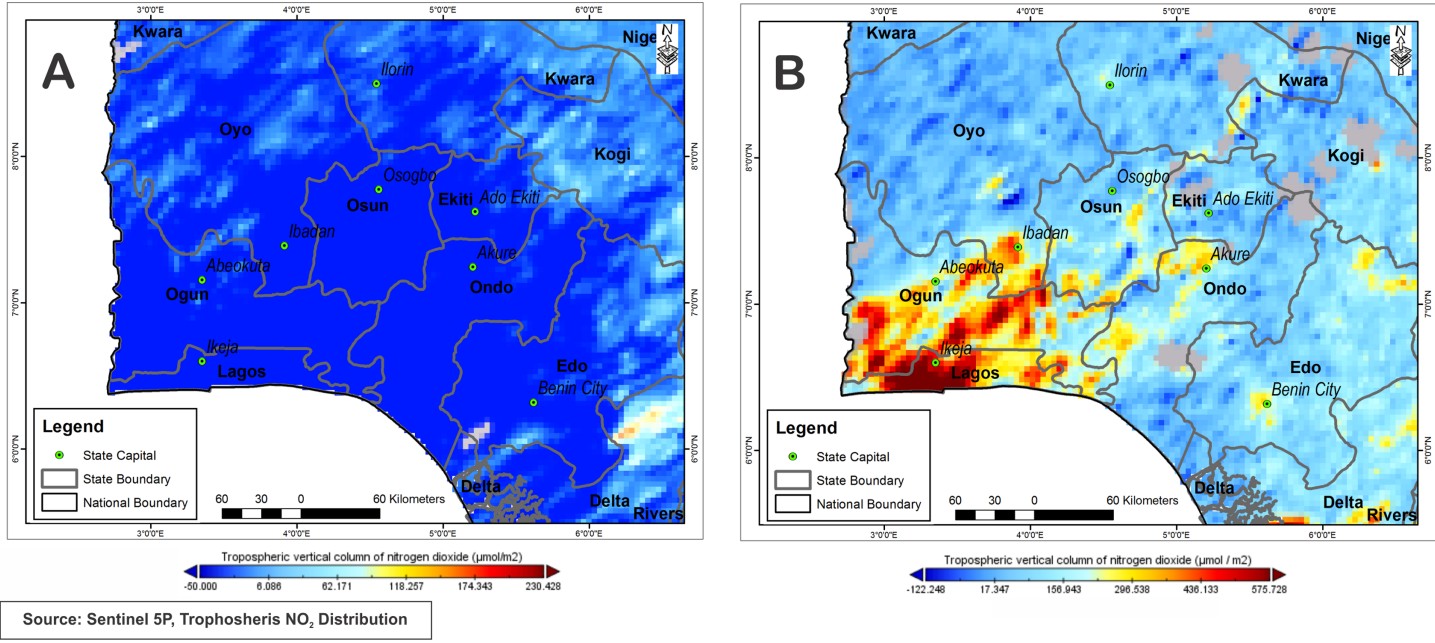

**Figure 5** Nitrogen dioxide concentration levels over Lagos and Ibadan during lockdown period April, 10–18, 2020 (A) and pre-lockdown period Dec., 10–14, 2019 (B). Full-size 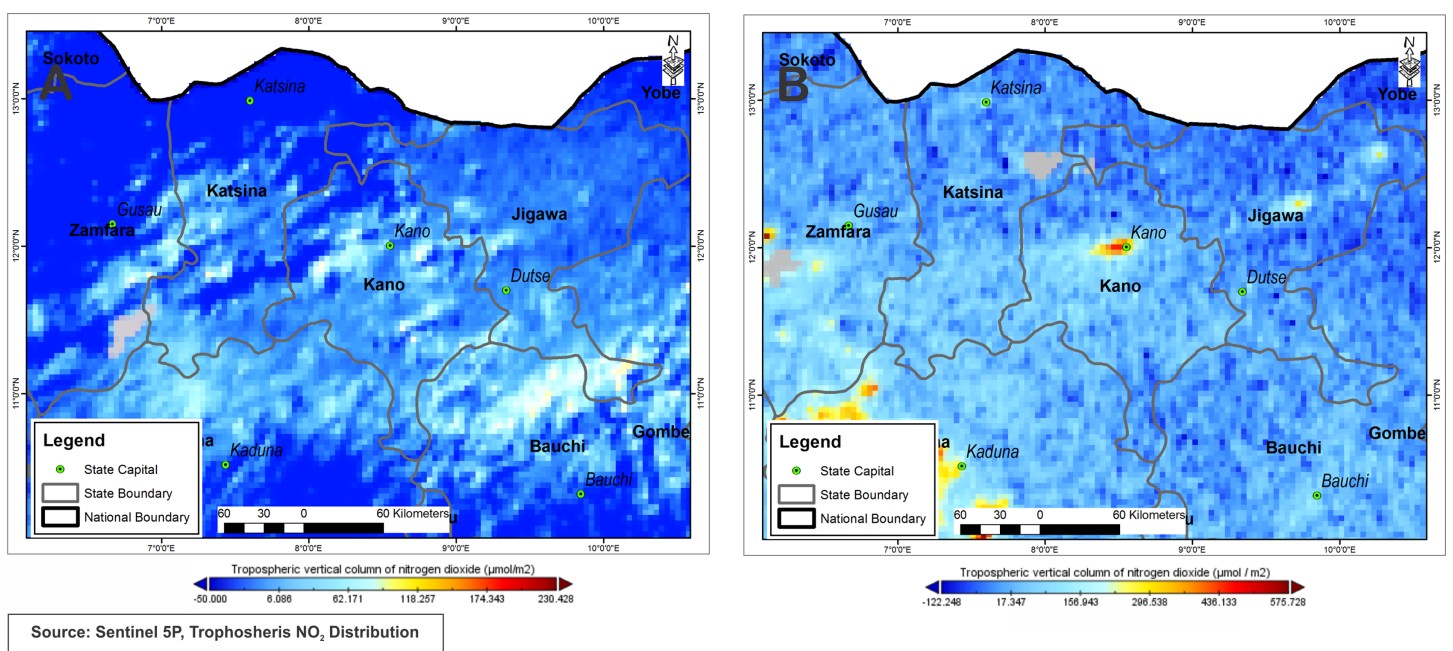 DOI: 10.7717/peerj.11387/fig-5

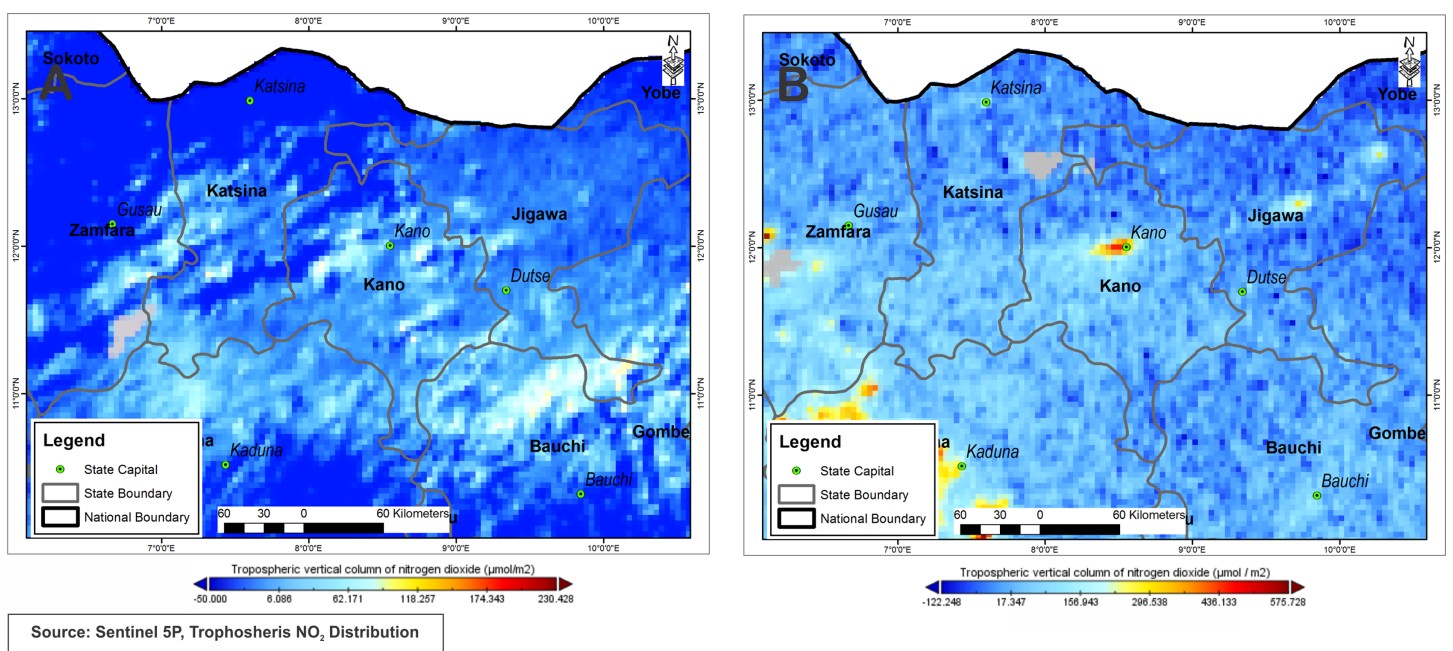

**Figure 6** Nitrogen dioxide concentration levels over Kano during lockdown period April, 10–18, 2020 (A) and pre-lockdown period Dec., 10–14, 2019 (B). Full-size 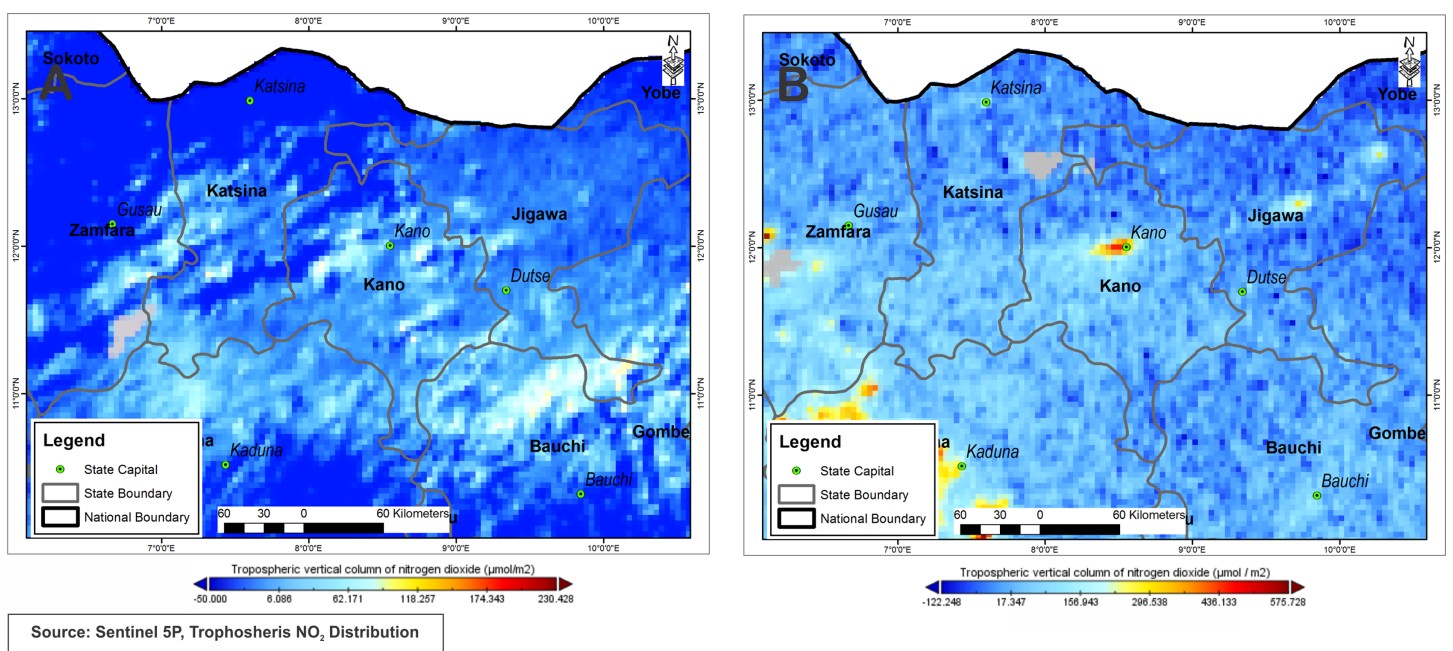 DOI: 10.7717/peerj.11387/fig-6

60.29 μmol/m$^2$ within Anambra state. It is noted that the pollution have fallen further in the area with the persistence of a few spots with high NO$_2$ concentrations, which is normal since this city is one of the most polluted in terms of air pollution in the world.
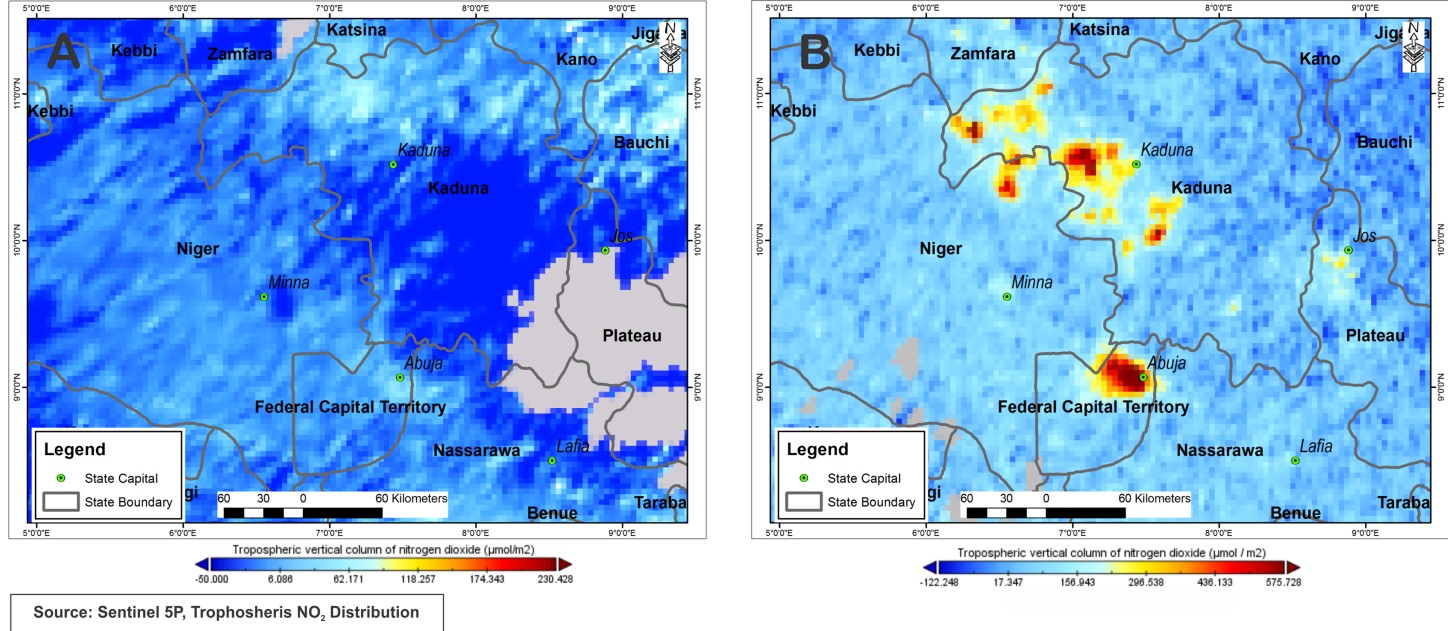

**Figure 7** Nitrogen dioxide concentration levels over Abuja during lockdown period April, 10–18, 2020 (A) and pre-lockdown period Dec., 10–14, 2019 (B).

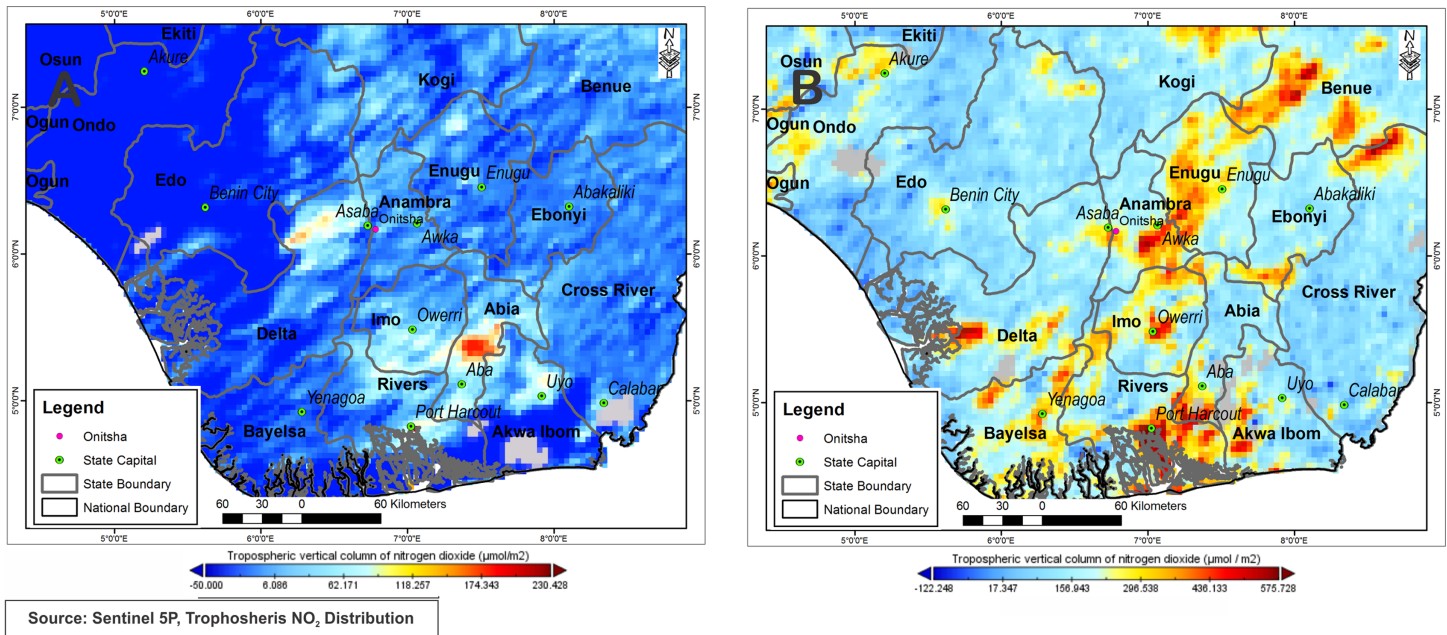

**Figure 8** Nitrogen dioxide concentration levels over Owerri and Onitsha during lockdown period April, 10–18, 2020 (A) and pre-lockdown period Dec., 10–14, 2019 (B).

## DISCUSSION

COVID-19 is a major pandemic and global issue that caused severe health, social and economic impacts in every country. Nevertheless, Africa has been reported to have had less

impact due to its early measures taken to abate this disease due to its recent experiences in similar infectious disease over the past decades.

This study analyzed the impact of COVID-19 on $NO_2$ concentration in six major cities in Nigeria; namely: Lagos, Ibadan, Kano, Abuja, Owerri and Onitsha. To understand the impacts of the lockdown conditions, Sentinel 5P Near Real Time level 2 Nitrogen dioxide ($NO_2$) satellite images extracted for specific durations in 2 separate months (December, 2019 and April, 2020) as shown in Figs. 5–8. The results show significant reduction in $NO_2$ concentrations over the cities under investigation due to the lockdown measures taken by the government. These reductions in $NO_2$ could be due to restrictions on sources of anthropogenic activities such as transportation and commercial activities. Moreover, the observations related to the dramatic reduction in air polluting emissions in Nigeria are in agreement with what has been reported for other regions globally. It is worth noting that $NO_2$ emissions can be affected by seasonality. However, more data is required to cover the same periods to avoid the influence of seasonality changes between winter and spring. Recently, *Fuwape, Okpalaonwuka & Ogunjo (2020)* recently assessed the impact of COVID-19 lockdown on air quality in three locations within Nigeria; in doing these, they compared historical air pollution (2005–2019) and air pollution during COVID-19 lockdown (especially for the months of January to April, 2020) to determine the level of air quality change using boxplots. Their results revealed a reduction of 1.1, 3.0, and 21.8% change in $NO_2$ levels for Lagos, Kaduna and Port Harcourt; respectively during the period of the lockdown. In addition Lagos and Kaduna saw an increase of 54 and 10% in $SO_2$ levels respectively during the same period. There was also a reduction of $SO_2$ levels by 37% in Port Harcourt during the lockdown period. They attributed this differences between 2005–2019 (before lockdown period) and January to April, 2020 (during lockdown period) to be due to different levels of enforcement of the lockdown in the different study locations. Further detailed maps of $NO_2$ levels in Wuhan, China from the 1st of January through the 25th of February of last year (2019) show the region covered with intense dark layers indicating much higher concentrations of $NO_2$. The images were compared with the same period in 2020 after the breakout of coronavirus where reduced levels of $NO_2$ have been observed (*Isaifan, 2020*).

Recently, *Bashir et al. (2020)* studied the correlation between the fast spread of COVID-19 and the climate indicators such as temperature and humidity in New York City in USA (*Bashir et al., 2020*). They found a correlation between the minimum and average temperatures with the spread of COVID-19 indicating that higher temperature serves as a driver for the COVID-19. Moreover, humidity was found to be another factor for the spread of COVID-19 as it contributed towards the rapid transmission of COVID-19 within New York City. In addition, the population density (8.54 million people, 26,403 people per square mile) and being a cultural and financial capital of the world, New York City experiences large amount of global travel all of which plays a role in rapid transmission of Covid-19 (*Bashir et al., 2020*). Similarly, the higher densely populated cities in Nigeria such as Lagos had extremely higher levels of pollution, and hence more confirmed cases and deaths are expected.

## CONCLUSIONS

In this study, the variation of $NO_2$ which was collected from the satellite (Sentinel-5P) was used to indicate the significant reduction in the level of $NO_2$ in six major cities in Nigeria. The observations compare the level of $NO_2$ emissions before and after lockdowns. Two main durations were compared (December, 2019 and April, 2020). $NO_2$ emission in the various cities under study shows a significant drop after the first coronavirus was announced in Nigeria, this could be due to restriction of vehicular and commercial activities as also reported in various cities across the world. There could be an element of uncertainty in the results due to seasonality, as the comparison is done with a different season. However, the magnitude of change due to lockdown is probably much higher than the seasonal variability. The outcomes indicated that the novel coronavirus is considered a blessing in disguise. The current status of air quality in terms of lower $NO_2$ concentrations has significantly improved. The impact might be temporary but there is a very good opportunity for all nations to learn about the outcomes of lockdown with minimum impact on economy.

## ACKNOWLEDGEMENTS

We appreciate the World Health Organization (WHO) and the Nigerian Centre for Disease Control (NCDC) for granting us the opportunity to use some of its publications.

### Funding

The authors received no funding for this work.

### Competing Interests

The authors declare that they have no competing interests.

### Author Contributions

- Johnson Adedeji Olusola conceived and designed the experiments, performed the experiments, analyzed the data, prepared some figures and/or tables, authored or reviewed drafts of the paper, and approved the final draft.
- Adebola Adekunle Shote conceived and designed the experiments, prepared main figures, authored or reviewed drafts of the paper, and approved the final draft.
- Abdellah Ouigmane authored or reviewed drafts of the paper, and approved the final draft.
- Rima J. Isaifan analyzed the data, authored or reviewed drafts of the paper, and approved the final draft.

### Data Availability

The raw data used to generate the figures and tables are available at the following sites:

- https://www.ncdc.gov.ng/diseases/sitreps/?cat=14&name

- https://covid19.ncdc.gov.ng/
- Situation report - 142, Coronavirus disease 2019 (COVID-19),
10 June 2020: https://www.who.int/emergencies/diseases/novel-coronavirus-2019/situation-reports.

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
