# Peer review of "The impact of COVID-19 pandemic on nitrogen dioxide levels in Nigeria"

_PeerJ, doi:10.7717/peerj.11387_

## Round 0.1 · original submission · Major Revisions

The authors are encouraged to sincerely incorporate all the suggestions mentioned by the referees which will make it a good informative paper. You must address COVID impact in detail which is lacking.

·

Basic reporting

• The manuscript is well written but there are areas of improvement (grammatical errors and sentence re-structuring). Kindly review the complete manuscript for improving its quality to ensure that an international audience can clearly understand. Some of the examples are given below:

Line 46: edit (after it originated)
Line 48: edit (measure that was termed)
Line 53: edit ((December 2019/April 2020/results show a significant)
Line 54: edit (during the lockdown/during the pre-lockdown)
Line 71: edit (and is declared)
Lines 102-103: edit (humans as it poses)
Line 113: edit (advent of coronavirus)
Line 129: edit (he Nigerias' air quality)
Lines 144-145: reference needed (In addition, these cities are among the top major inhabited cities in Nigeria and among the most polluted)
Line 147: reference for spatial resolution (spatial resolution of 5.5 km)


2) Although the resolution of figures meets the basic requirements for submission but it is not good for readers and details are hidden due to poor quality.

3) Figure-2: What is disease outbreak status here (number of positive cases or cases per million population or else)

4) Figures-5,6,7,8: there is no legend to interpret whether red is high or blue is high in concentration.

5) Main data for NO2 concentration is missing (can be provided as a table too as zonal statistics for each state that was studied).

Experimental design

1) I feel that intermittently the paper loses its focus between impact of COVID-19 on NO2 concentration vs its impact on general population.

2)Although the authors have utilized satellite imagery to do a comparative analysis but they haven't justified the use of satellite image versus the actual ground monitoring stations data. Authors need to cite papers that correlate these two sources of NO2 pollution assessment.

3) Comparing NO2 concentrations for December 2019 and April 2020 may not be the best way to proceed as many air pollutants show a trend i.e. they vary between winter, summer, rainy seasons due to several factors and atmospheric phenomenon. It is best to compare apples to apples i.e. baseline data should have been April 2019 and not December 2019.

Validity of the findings

1)The manuscript in its current form needs major re-structuring including re-running some analysis, improving figures, providing data, refining the introduction and discussion section to emphasize mainly on NO2 pollution rather than COVID-19 cases in general.

2) Authors have just mentioned that there is a significant improvement in NO2 pollution which is based on just two months (unaccounting for seasonality). There is no mention about how much percent improvement was achieved for this period.

Additional comments

Its a good topic and good start but it needs some improvement before others can make use of this information. Best wishes!

Reviewer 2 ·

Basic reporting

The paper poses an interesting and timely question. The writing, however, needs work. Ample suggestions, and most of my review, are shown in the attached review.

Experimental design

The manuscript states that the study "...compares the levels of NO2 density before the advents of the first reported case of COVID-19 in Nigeria and its levels during the lockdown in selected cities to assess its impact on the air quality of Nigeria." Unfortunately the authors simply provide satellite images that represent NO2 concentration over four cities/regions in December 2019 and again in April 2020. The authors did not suggest, nor did they offer any analyses of these images (which, don't even have a legend to suggest what the concentration levels were); their study simply provided a brief visual assessment of the before and after images.

Validity of the findings

The authors repeatedly suggest a "significant decrease", "significant drop" or more troubling, "NO2 pollution has disappeared" - they have offered no technical or statistical assessment to justify their findings.

Additional comments

The title of the paper suggests an analysis of changes in NO2 concentrations over Nigeria before and after the COVID-19 lockdown, but there was little information provided on this question. Most of the discussion provided in the Results was in fact background information on the cities where the authors looked at satellite images. The background information should have been presented in the introduction, but that would have magnified how little NO2 analyses were provided in the Results.

The study does represent an interesting and useful question, but this manuscript really only provides brief discussions on the visual changes observed between the before and after images, which does not provide much scientific information.

To make this study meaningful to the PeerJ community, the authors need to perform enough technical analyses (e.g. statistical analyses of the NO2 data before and after the lockdown) to justify any statements about changes in NO2 concentrations. All information about infection/death rates or economic impacts in each region studied need to be moved out of "Results" and into "Introduction", and the "Results" section should only focus on NO2 data.

Annotated reviews are not available for download in order to protect the identity of reviewers who chose to remain anonymous.

---

## Round 0.2 · Minor Revisions

The revision has been found suitable. Please do include some of the answers to the comments in text at appropriate places. Once you can incorporate these changes, your manuscript may be acceptable.

·

Basic reporting

There is an overall improvement in the manuscript language, structure and figures.

Experimental design

I believe the authors have done a good job in improving the manuscript for international readers.

The only minor thing that I will suggest adding to the manuscript before publishing it is a better justification of comparing NO2 concentrations for December 2019 and April 2020. No doubt that COVID-19 lockdown has reduced pollution but the seasonality factor needs to be accounted for as well at least when we are not comparing the same time periods. This can be done by simply citing recent articles or comparing a trend during the same period for the previous year. Google Earth engine apps can be helpful as well.

Validity of the findings

The topic is good, results are valid and it is good to be published with very minor edits as mentioned in the previous section.

Additional comments

Good work in improving the script. One last improvement to add and it is all set - best wishes!

---

## Round 0.3 · Minor Revisions

The rebuttal provided by you is not fully satisfactory. However, your paper may be acceptable but you need to put the following lines in your abstract as well as in the conclusion:

“There could be an element of uncertainty in the results due to seasonality, as the comparison is done with a different season. However, the magnitude of change due to lockdown is probably much higher than the seasonal variability.”

---

## Round 0.4 · accepted · Accept

Authors have satisfactorily incorporated suggested changes and hence paper is now ready for publication.